# The Migration and the Fate of Dental Pulp Stem Cells

**DOI:** 10.3390/biology12050742

**Published:** 2023-05-19

**Authors:** Nadia Lampiasi

**Affiliations:** Istituto per la Ricerca e l’Innovazione Biomedica, Consiglio Nazionale delle Ricerche, Via Ugo La Malfa 153, 90146 Palermo, Italy; nadia.lampiasi@irib.cnr.it; Tel.: +39-091-680-9513; Fax: +39-091-689-5548

**Keywords:** hDPSCs, PIEZO1, ATP, migracytosis, blebbing, YODA1, leader and follower, mechanotransduction

## Abstract

**Simple Summary:**

The importance of stem cells for regenerative medicine has grown significantly in recent years. This is because stem cells can differentiate into multiple cell types and are often easy to recover. Dental pulp stem cells can differentiate into odontoblasts (dentin), osteoblasts, chondrocytes, adipocytes and nerve cells. They are easy to recover and can proliferate, migrate and differentiate in vitro. The regeneration of damaged tissue depends on the homing of the recruited cells and thus on cell migration. However, not all stem cells are equally capable of migrating. Indeed, they may use different modalities, different times or different stimuli. Amoeboid and mesenchymal migration are commonly utilized by mesenchymal stem cells to move, including dental pulp stem cells. Recently, migracytosis and dynamic blebs also appear to be two modalities used by mesenchymal stem cells, although there is still no experimental evidence for their use in dental pulp stem cells. Cells move in response to environmental stimuli interacting with specialized cell receptors. Environmental stimuli can be of a different nature: chemical or physical, including mechanical, which depends on forces that interact with the cells. This review aims to shed light on the characteristics used by dental pulp stem cells to migrate in relation to differentiation options.

**Abstract:**

Human dental pulp stem cells (hDPSCs) are adult mesenchymal stem cells (MSCs) obtained from dental pulp and derived from the neural crest. They can differentiate into odontoblasts, osteoblasts, chondrocytes, adipocytes and nerve cells, and they play a role in tissue repair and regeneration. In fact, DPSCs, depending on the microenvironmental signals, can differentiate into odontoblasts and regenerate dentin or, when transplanted, replace/repair damaged neurons. Cell homing depends on recruitment and migration, and it is more effective and safer than cell transplantation. However, the main limitations of cell homing are the poor cell migration of MSCs and the limited information we have on the regulatory mechanism of the direct differentiation of MSCs. Different isolation methods used to recover DPSCs can yield different cell types. To date, most studies on DPSCs use the enzymatic isolation method, which prevents direct observation of cell migration. Instead, the explant method allows for the observation of single cells that can migrate at two different times and, therefore, could have different fates, for example, differentiation and self-renewal. DPSCs use mesenchymal and amoeboid migration modes with the formation of lamellipodia, filopodia and blebs, depending on the biochemical and biophysical signals of the microenvironment. Here, we present current knowledge on the possible intriguing role of cell migration, with particular attention to microenvironmental cues and mechanosensing properties, in the fate of DPSCs.

## 1. Introduction

Adult stem cells are a resource for living organisms that allow for the repair and/or regeneration of damaged tissues. In general, there are two different approaches to regenerate damaged tissue using stem cells: cell homing and cell transplantation, both of which imply cell migration. Mesenchymal stem cells (MSCs) are found mainly in the bone marrow (BM), but also in the adipose tissue and in the pulp of the tooth. They can give rise to osteoblasts, chondrocytes and adipocytes. Dental pulp stem cells (DPSCs) are unique because they arise from the ectomesodermal embryonic tissue that forms the neural crest. For this reason, in addition to the cell types described above, they can give rise to odontoblasts (specialized osteoblasts) and nerve cells (astrocytes, glia cells and oligodendrocytes). The microenvironment in which stem cells are found affects their differentiation. In the case of naïve DPSCs, the niches in which they are located are innervated, supplied with blood and inside a rigid structure (tooth). The decision between the renewal and migration/differentiation of DPSCs depends on their interactions with stromal cells and ECMs in the niches. Different niches possess different DPSCs. As an example, after tooth injury in the apical part of the pulp, there was a population of highly proliferative potential which was Notch2-positive [1], whereas in the perivascular niches, DPSCs were positive for Oct3/4 stemness markers [2]. It has been shown that in vivo human (h) DPSCs migrate and can repair dentin by regenerating damaged odontoblasts in the tooth [3]. Furthermore, hDPSCs transplanted into mice can promote bone regeneration in defective calvaria [4] or migrate to ischemic areas, i.e., areas of cerebral infarction, and express specific neural markers [5]. Additionally, when induced in vitro as neural cells, they can differentiate in vivo into mature neurons or astrocytes [6]. However, hDPSCs can also repair damaged nerve tissue through paracrine mechanisms involving chemotaxis and the proliferation of endogenous neural stem cells (NSCs) [7], or they can reduce ischemic damage through the inhibition of microglial activation and the expression of pro-inflammatory cytokines [8]. When comparing MSC transplantation with the practice of homing for tissue regeneration in preclinical animal models, the latter is safer and more effective [9]. However, the main limitations are the poor cell migration of MSCs and the limited information we have on the regulatory mechanism of the direct differentiation of MSCs.

In this review, we describe the migration modes and time used by hDPSCs in response to different microenvironmental cues, focusing on the different cell fates, especially the specification of odontoblasts/osteoblasts and/or nerve cells, where reported.

## 2. HDPSCs

HDPSCs can be purified from dental pulp essentially by two methods. The first is based on the enzymatic digestion of the dental pulp with collagenase and dispase and/or trypsin [10] (Table 1).

The harvested cells were dispersed in the medium, plated and left to proliferate. In turn, they formed colonies of heterogeneous cells, including stem cells from various niches in a mixture, and epithelial cells, stromal cells, perivascular cells, etc. [19] (Figure 1A). In this regard, heterogeneous cells expanded in a serum-free medium produced two DPSC subtypes, those being adherent (ADH) and non-adherent (non-ADH) populations according to their differential adhesion to plastic; however, both populations displayed osteogenic and neurogenic differentiation [23,26]. The second method consisted of putting the pulp, or fragments, directly in a plate with a culture medium and waiting for the DPSCs to come out after about 10–15 days (explant method) (Table 1, Figure 1B and Figure 2A) [16]. The tissue piece is present during the primary culture and therefore, DPSCs reside in the dental niches with stromal cells and extracellular matrix (ECM), since no proteolytic enzymes are added in the culture. DPSCs take a long time to come out from the pulp, but they take advantage of the presence of other cells and the ECM. This method makes it possible to observe the cells that are induced to migrate. Indeed, DPSCs gradually emerge from the tissue, whereas non-migrating cells remain inside the tissue and can migrate later (for agreement with DPSCs residing in different niches, see [1], or if they are not stem cells, they undergo apoptosis [1,2,49]). However, DPSCs obtained by the explant method produce a more homogeneous population (i.e., subsequent waves, see [20]), and unattached and adherent cells, different from DPSCs, are present in the culture [24]. However, unattached cells will be gradually removed after refreshing the culture media, and adherent cells are unable to survive/proliferate and will be lost during the first few subcultures [20,49].

Therefore, the two isolation methods yielded different subpopulations of cells, even though regardless of the recovery method, DPSCs showed the same trilineage differentiation potential when not pre-selecting for specific marker expressions [24,27,50]. In general, in vitro hDPSCs are most likely induced to migrate and proliferate following environmental cues such as chemical and biophysical stimuli, for example, changes in stiffness and rigidity (both are mechanical stimuli) [51,52]. However, in the explant method, a wound healing response is triggered due to the production of cytokines and factors released by the injured tissue, which promote migration [31,49]. Cells harvested with the explant method migrate as “leaders” and “followers”, where leaders migrate first as single cells [29,53] and guide the migration and followers follow the guide, connoting a subdivision of the group into distinct fractions [54].

Many mathematical models (stochastic models) have been developed to explain the various types of cell migration. However, these models are often based on the migration of clustered cells (tumors) or activated lymphocytes (taxis), whereas DPSCs, as said before, can migrate as single cells [55]. In addition, some models are based on the idea that the nucleus occupies a central position, but this is not always true. For example, in DPSCs, the nucleus is often lateral or in the back. Moreover, in response to environmental cues, many cells have the capacity to turn off their default migration mode from mesenchymal to ameboid and vice versa [56]. Another important feature to take into consideration is given by the stimuli that recruit the “leaders”, which is not fully understood and can be single or double. To date, the literature is still scarce concerning mathematical models that explain cell migration in the presence of two stimuli. The first cue concerns the choice of direction and the second, usually of mechanical origin, concerns the speed that the cell can reach going in that direction [57,58,59]. Speed is important because the fastest cells (leader) can lead the others (follower). However, further studies are needed to define a good mathematical model that accounts for the migration of DPSCs.

## 3. Migration

Migration is an essential activity for MSCs to reach damaged sites and contribute to repair and differentiation. The migration of single cells occurs essentially by mesenchymal and amoeboid movement and numerous studies have been done by analyzing in vitro cell migration on different components of the ECM as well as on other biomaterials [60]. However, cell migration depends on various microenvironmental cues such as soluble factors (chemotaxis) [61], substrate-related factors (haptotaxis) [62,63,64] and mechanical signals (durotaxis) [65,66].

Mechanistically, migration and adhesion depend on the presence of a network consisting essentially of two proteins: actin and myosin (non-muscular). In general, actin is organized in an array which depends on its ability to assemble and disassemble the domains according to different contexts and environmental signals. Indeed, actin domains are organized in filaments that give rise to plasma membrane protrusions known as lamellipodia, filopodia and blebs [67]. Instead, the myosin is organized in bipolar filaments, which slide over the actin array and generate the forces necessary to move the cells [68]. In addition, a thin mesh of actin, myosin filaments and associated proteins is found under the plasma membrane, forming a cortex which contributes to the changes in the shape of the cells themselves [69].

### 3.1. Lamellipodia

Lamellipodia are dependent on the generation of a branched Arp2/3 complex, which produces forces, thus overcoming membrane tension and driving protrusion. They are frequently associated with mesenchymal migration, for example, in fibroblasts, hematopoietic cells, innate immunity cells, tumor cells, embryonal stem cells and DPSCs (Figure 2B,C), both in 2D and 3D (named ruffles) and also in vivo [70,71].

In addition, the lamellipodium is the site for most cell–matrix adhesions [72], formed through integrins binding to ECM proteins and subsequent clustering in focal complexes or focal contacts at the distal margin of the lamellipodium. It is noteworthy that some focal complexes mature into focal adhesions (FAs) that are connected to bundled actin stress fibers (SFs) (see SF paragraph below) (Figure 2D) [68,73]. Interestingly, DPSCs use lamellipodia to migrate [29] independently of substrate stiffness (Figure 2E). As stiffness decreases, FA decreases and migration rate increases [33].

### 3.2. Filopodia

Filopodia, unlike lamellipodia, are independent of the Arp2/3 complex. They are protrusions of the cytoplasmic membrane filled with actin, which polymerize, generating forces towards the membrane which rapidly extend and retract the protrusions [74,75] (Figure 2F). In general, cells can sense the surrounding environment by using filopodia and this includes sensing other cells and ECM to adhere or interact with them [76]. They can also serve as bridges between cells for the transport of various cargoes [35,77,78,79]. Many studies reported the presence of filopodia/lamellipodia in hDPSCs [29], evenly cultured over a porous surface and used for migration [76].

### 3.3. Stress Fibers

In animal cells, the actin cortex is composed of very ordered actin–myosin structures called SFs. SFs are used by cells that migrate according to the mesenchymal modality, while these fibers are not evident in amoeboid cells [79,80]. In migrating cells, SFs are found mostly in the back of cells and under or near the nucleus, contributing to tail retraction during migration [80,81]. Indeed, the nucleus of migrating polarized mesenchymal cells is in the back of the cell, pushed by forces dependent on myosin activity [82,83]. SFs are present in DPSCs [30], in association with FAs, especially in cells plated on a stiff matrix (plastic wells), or following the application of forces, representing an important mechanosensitive mechanism for cells [84,85].

### 3.4. Blebbing

Blebs are spherical swellings of the plasma membrane formed by amoeboid cells that show plasticity and a high degree of deformability. Amoeboid migration does not require active F-actin polymerization, but depends on myosin II-mediated contractility, which generates hydrostatic pressure against the plasma membrane, producing blebs [68,86,87] (Figure 2G, Appendix A). Their formation can be induced by numerous mechanical and chemical stimuli, including changes in external stiffness and osmolarity. Amoeboid motility is widely used, including, for example, use by human embryonal stem cells (hESCs). In this case, blebbing is driven by ROCK–myosin activity [88] and depends on the substrate and attachment of cells through an integrin-FAK pathway [89]. However, little is known about the upstream signaling of dynamic blebbing. Many studies underrepresent the role of purinergic P2 receptors (see Mechanosensing and Mechanotransduction paragraph) in cell types, including hepatocytes [90] thymocytes [91], macrophage cell lines [92] and hESCs [89]. To our knowledge, there are no data on the phenomenon of dynamic blebbing in DPSCs.

However, these three types of protrusions (lamellipodia, filopodia, blebs), which generate different types of migration (mesenchymal and ameboid), are interconnected. Indeed, it has been demonstrated that interfering with lamellipodia formation promotes plasma membrane bleb generation [68].

### 3.5. Migrasomes

Migrasomes have been discovered very recently in many cell types, including macrophages, primary neurons, ESCs and in the circulating blood. They are organelles produced by migrating cells which depend on the polymerization of actin and are positive for tetraspanin 4 (Tspann4) [93]. As the cells migrate, long, thin membrane projections called retraction fibers (RFs) are left in their wake at the rear of the cell [94]. Migrasomes have been highlighted on these fibers, taking on the appearance of a pomegranate-type structure with an oval organelle with diameters from 0.5 µm to 3 µm and cytosolic contents, which include proteins without a signal peptide [93] (Figure 2H). Eventually, the retraction fibers break up and migrasomes are released into the medium or directly taken up by surrounding cells.

The primary function of migracytosis is probably cell-cell communication. The great importance of migracytosis is given by the fact that the cells releasing migrasomes (outgoing) give spatial and biochemical information that can be acquired by the following cells (incoming) [95,96]. Indeed, it has been reported that migrasomes could play a role in cellular chemotaxis [95]. In all cases, the migrasomes left along the way are endocytosed by the recipient cells and modify the cells [97]. However, the question arises as to why these organelles are not degraded with their contents by the lysosomes of the recipient cells. To date, there is no clear answer. One hypothesis is that a similar mechanism occurs during DNA transfection, but this is unclear [96].

Eventually, migrasome formation depends on cell migration, and they are known to be used by migrating cells as signals for other cells. The DPSC “leader” subtype could use this mechanism to signal the “follower” subtype on which direction to take, as it has been shown that more migrasomes are released from faster and more persistent cells [98]. However, further studies are needed to better elucidate the process and the possibility that DPSCs use this type of signaling to direct migration in vitro and in vivo.

## 4. Mechanosensing and Mechanotransduction

Since stem cells are exposed to mechanical forces, they have developed many ways to adapt and protect themselves from the mechanical challenges they continually experience in environmental niches [99]. Mechanosensing and mechanotransduction are mechanisms by which cells sense the extracellular environment, mechanical stimuli in particular, and convert into intracellular biological signals. Several receptors are present on the cell membrane with the function of transducing mechanical signals, including mechanosensitive ion channels such as PIEZO, G protein-coupled receptors (GPCRs) and integrins [100,101,102]. These receptors can interact directly or indirectly with microtubules and/or actin filaments inside the cell by modifying their structure [103]. Several classical signaling pathways can transduce mechanical signals to biochemical input, including mitogen-activated protein kinases (MAPKs) [11]. These biochemical cascades promote transcription factor activation and transcriptomic changes, which are crucial for stem cell fate decisions [104]. As a result, cells can remodel their cortical cytoskeleton and cell membrane, adapting to the mechanical forces they are subjected to until they encounter new stimuli [105]. In this way, cells can develop processes that allow them to grow, proliferate and migrate to survive and protect themselves from excessive mechanical force [106].

Among the main types of forces, there are those external to cells and those “outside-in”, for example, fluid flow-induced shear stress, osmotic stress and pressure-induced membrane stretch [107]. Other types of mechanical forces are those generated by actin–myosin traction in FA zones, i.e., “inside-out”, which is used by cells to examine the mechanical and spatial properties of the ECM, move cargo inside the cells, determine cell shape and support cell substrates [107,108].

Mechanistically, a recent study demonstrated that YODA1 (ion channel-specific activator) activated PIEZO1 and stimulated hDPSC migration in vitro through the noncytolytic release of ATP [12] and its downstream signaling pathways (MEK/ERK, p38 MAPKs) [109,110]. Many other studies underrepresent the role of PIEZO1-ATP in DPSC migration [87,111], proliferation and differentiation through P2 purinergic receptors (P2X ionotropic and P2Y metabotropic receptors) [14,37,38,112]. Transcriptional coactivators, yes-associated protein (YAP) and transcriptional coactivator with PDZ-binding motif (YAP/TAZ), are identified as mechanotransducing transcription factors. They respond to a series of mechanical stimuli (stiffness, topology and stretching of the ECM), promoting genes and proteins expression when translocated into the nucleus [113], as in the case of YAP/TAZ-regulated genes, CTGF and ANKRD1 [39], and OC, OPN and BSP in DPSCs [114].

## 5. The hDPSC Fate

Self-renewal and differentiation are stem cell choices that need to be considered to define a good DPSC migration model. These two characteristics may depend on environmental cues where the cells grow (niches, in vitro substrates) and on some correlated factors, including the stiffness or composition of the ECM and materials, the presence of cytokines, chemokines, hormones and growth factors, the concentration of O_2_ and of course, mechanical stress forces. Indeed, DPSCs can migrate, proliferate and/or differentiate into multiple cell lineages under the influence of the biophysical and biochemical properties of the microenvironment.

Substrate stiffness can modulate cell morphology, adhesion, migration and differentiation through cytoskeleton arrangement. Topographic cues can be provided by differences in stiffness, roughness and the pore size of scaffolds produced in the laboratory. Several studies have investigated this matter and found a relationship to changes in cell morphology and movement that depend on the actin’s network. In addition, these changes can influence stem cell behavior by promoting the preferential osteogenic/odontogenic differentiation of DPSCs [13] and adipose stem cells (ASCs) [115]. Usually, substrates with different rigidity prompt a preferential direction of migration from a stiff to soft matrix. However, DPSCs can unexpectedly move from the soft to the stiff substrate and vice versa without any preferential direction, independent of the myosin II activity and of YAP nuclear translocation, through the activity of lamellipodia and mesenchymal migration [33]. Therefore, in the absence of factors that can recruit the DPSCs in vivo, such as those released during dentin damage or pulp inflammation, the stem cells are free to move randomly without a precise direction but retain the ability to differentiate into odontoblasts. Magnetic materials are used in dental clinics and provoke a static magnetic field (SMF). This stimulus can provoke the rearrangement of a DPSC cytoskeleton, promoting DPSC migration and proliferation through MMP-1, MMP-2 and FGF-2, TGF-β and VEGF gene expression [13,39]. An interesting study analyzed the effect of SMF on migration, proliferation and differentiation in DPSCs obtained with the explant method [29]. The results showed that the SMF-treated group moved by mesenchymal collective cell migration behavior, whilst the sham-exposed groups moved mainly by single-cell migration and with a random direction. Both groups differentiated in odontoblasts even though the SMF-treated group of DPSCs did so more efficiently [29].

The ECM varies greatly in composition depending on the microenvironment; therefore, many studies have been conducted using the hydrogel as a matrix with different coated ECM proteins used to evaluate the migration. The hydrogels containing hyaluronic acid (HA) influence the chondrogenesis differentiation of MSCs [116], whereas hydrogel-containing gelatin and fibrin enhanced cell migration and induced the odontogenic differentiation of DPSCs [17].

Emblematic of factors eventually present on the ECM is the example of stromal cell-derived factor 1 (SDF1), which is a chemokine released by injured tissues as brain, hearth and bone. In vitro studies showed that SDF1 improved DPSC migration [15,40] and in combination with other chemo-attractants such as bone morphogenetic protein 7 (BMP-7), parathyroid hormone (PTH) and exendin-4 (EX-4), improved the efficiency of the osteogenic differentiation of DPSC and periodontal ligament stem cells (PDLSC) [18,41,42,117]. Other growth factors, such as glial cell-derived neurotrophic factor (GDNF) [22] and insulin-like growth factor-binding protein 5 (IGFBP5), promoted DPSC migration and enhanced the osteo/odontogenic and neurogenic differentiation of DPSCs [43]. Injured tissues can secrete the neuropeptide substance P (SP), which may exert immunomodulatory and stem cell recruitment roles [118,119]. Very recently, SP has been implicated in the odontoblastic differentiation of DPSCs during the reparative genesis of dentin. In addition, the SP/NK1R signaling pathway, expressed in DPSCs, plays a fundamental role in the regulation of cell recruitment upon damage, suggesting a role as a migration/differentiation inducer [32].

The inflammatory microenvironment, which is generated in the dental pulp because of an insult (trauma or infection), represents an essential prerequisite for tissue healing and regeneration and can influence the fate of DPSCs. Indeed, it has been reported that Pentraxin-3 (PTX3), which is an inflammatory mediator, is involved in the migration and osteogenic/odontogenic differentiation of hDPSCs [25]. The chemokine receptor CXCR3 (CD183) was found on 30% of the adherent (ADH)-DPSCs [23], associated with neuroinflammatory responses and their potential involvement in homing neural progenitors to sites of brain damage [120]. Another study showed that human-concentrated growth factors (hCGFs) alone and in combination with LPS increased DPSC migration and osteogenic differentiation [121]. A conditioned medium, such as human gingival fibroblast-conditioned medium (hGF-CM), increased the migration, proliferation, cell viability and odontogenic differentiation of DPSCs after H_2_O_2_ exposure [44].

Other interesting studies focused on the exosomes (Exos) and extracellular vesicles (EVs), which can cargo pivotal molecules for cellular recruitment and differentiation. Exosomes produced by undifferentiated or angiogenic differentiated hDPSCs (DPSC-Exos) contributed to the homing and angiogenesis of naïve DPSCs [34]. Other studies on Exos-DPSCs have used the enzymatic purification method to recover stem cells. In this regard, DPSC-Exos promoted cutaneous wound healing-related biological processes in mice, such as the positive regulation of cell motility, migration, proliferation, vasculature development and angiogenesis [44]. Exos obtained from DPSCs under hypoxic conditions (Hypo-Exos) demonstrated that LOXL2, an enzyme that catalyzes the crosslinking of elastin and collagen, is a key molecule mediating the angiogenic effect by promoting migration and proliferation [46]. EVs from Schwann cells (SCs) successfully increase the proliferation, migration and osteogenic differentiation of hDPSCs [47].

Little is known about the conditions that can induce preferential neural differentiation of DPSCs. Very recently, a study showed that a combination of EGF and bFGF are sufficient to induce neural phenotypic changes and biomarker expression on DPSCs, but these changes depend on proliferation rate [21] and ECM composition [36,48]. The damaged tissues release homing and growth factors, which can induce the recruitment of DPSCs from nearby sites. Indeed, in a model of neurodegeneration of hippocampal neurons, the upregulation of homing factors (SDF-1alpha, CXCR-4, VCAM-1, VLA-4, CD44, MMP-2) are induced in vitro and DPSC migration is increased [28]. Furthermore, in vivo systemic administration of DPSC/BM-MSC in an animal model with temporal lobe epilepsy (TLE) and an impaired blood-brain barrier (BBB), induced stem cell homing in the CNS and attenuated symptoms of neurodegeneration and neuroinflammation [28].

## 6. Conclusions

Based on what has been described in the previous paragraphs and our experimental observations, we want to suggest a hypothesis on the relationship between the migration, self-renewal and differentiation of DPSCs. We assume that the pulp of the tooth placed in culture without enzymatic digestion favors the leakage of stem cells, which sense the biochemical and mechanical stimuli represented by the injured tissue. These cells (leaders) are probably induced to migrate as single cells by the presence of two stimuli: one is represented by the release of factors following tissue damage (wound healing response) and the second is represented by the stiffness of substrate (plastic plate), which guides cell migration through their mechanosensing and mechanotransduction properties. Along the way, leaders can leave migrasomes, which can be engulfed by following cells (followers). Thus, topographic cues are obtained due to the content of the migrasomes, indicating a precise direction to move. The leaders proliferated and formed colonies far from the pulp, and they can potentially differentiate based on microenvironmental cues. After some time, other DPSCs emerge from the pulp, probably coming from other niches (with stemness properties) and containing a self-renewal capacity, which can be attracted by the signals released by the first cells that emerged (leaders). These cells do not migrate far, but remain close to the pulp and are induced to proliferate, suggesting a high self-renewal capacity. Surely, even these cells, if induced by appropriate environmental signals, would be able to differentiate into appropriately specialized cells. Eventually, in our opinion there are at least two different subtypes of DPSCs, likely residing in different niches: one capable of migrating and exploring the microenvironment in response to biochemical and mechanical stimuli, with the immediate ability to differentiate into specialized cells and repair damaged tissue; and another subtype that is able to proliferate and maintain a self-renewal capacity. However, more studies are needed to clarify these aspects of DPSCs.

## Figures and Tables

**Figure 1 biology-12-00742-f001:**
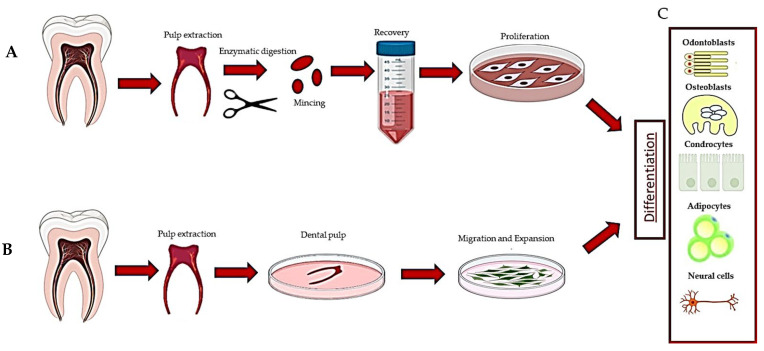
Description of the methods used to recover dental pulp stem cells: (**A**) digestion method, (**B**) explant method and (**C**) differentiation of DPSCs.

**Figure 2 biology-12-00742-f002:**
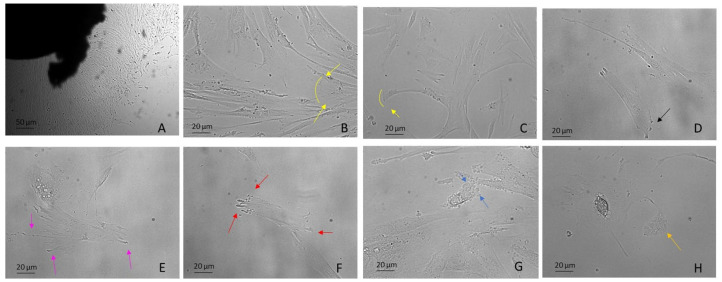
DPSCs harvested using the explant method from the third molar extracted for orthodontic reasons. Examples include: (**A**) DPSCs recovered by explant method (10×) (**B**,**C**) mesenchymal migration (20×), (**D**) FA (20×), (**E**) lamellipodia (20×), (**F**) filopodia (20×), (**G**) blebs and ameboid migration (20×), (**H**) migrasomes (20×). Light Microscopy (Zeiss Axio). DPSCs were isolated with the explant method from teeth removed for orthodontic reasons and the patients agreed to the use of their samples for research purposes. Bar = 20 µm; 50 µm.

**Table 1 biology-12-00742-t001:** Schematic list of the manuscripts cited in the review divided according to the DPSC recovery method.

Digestion Method	Explant Method	Not Specified	Cell Line
[3]	[1]	[8]	[11]
[5]	[2]	[12]	[13]
[6]	[7]	[14]	[15]
[10]	[16]	[17]	[18]
[19]	[20]	[21]	[22]
[23]	[24]		[25]
[26]	[27]		[28]
[20]	[29]		
[27]	[30]		
[31]	[32]		
[33]	[34]		
[35]	[36]		
[37]			
[38]			
[39]			
[40]			
[41]			
[42]			
[43]			
[44]			
[45]			
[46]			
[47]			
[48]			

## Data Availability

No data are available do to privacy.

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
