# Peer review of "The Migration and the Fate of Dental Pulp Stem Cells"

_biology, 2023, doi:10.3390/biology12050742_

Round 1

Reviewer 1 Report

The review provides a comprehensive overview of the migration and fate of dental pulp stem cells (DPSCs). It discusses the origin and characteristics of DPSCs, as well as their potential applications in regenerative medicine. The article also highlights the various factors that can influence DPSC migration and differentiation, such as growth factors, extracellular matrix, and mechanical stimuli.

Overall, the review article by Nadia Lampiasi provides valuable insights into the current state of research on DPSC migration and fate, and its potential therapeutic applications. The article is well-researched and well-written, making it a valuable resource for anyone interested in the field of regenerative medicine.

Only one mistake should be corrected

Page 7 line 267

O2 should be corrected with O2

Author Response

The review provides a comprehensive overview of the migration and fate of dental pulp stem cells (DPSCs). It discusses the origin and characteristics of DPSCs, as well as their potential applications in regenerative medicine. The article also highlights the various factors that can influence DPSC migration and differentiation, such as growth factors, extracellular matrix, and mechanical stimuli.

Overall, the review article by Nadia Lampiasi provides valuable insights into the current state of research on DPSC migration and fate, and its potential therapeutic applications. The article is well-researched and well-written, making it a valuable resource for anyone interested in the field of regenerative medicine.

  1. R) Only one mistake should be corrected Page 7 line 267 O2 should be corrected with O2
  2. A) I thank the reviewer for the appreciation. The mistake has been corrected.

Reviewer 2 Report

This review focuses on specific capabilities of DPSCs.

Albeit the number of reviews on these stem cells are in numerous, this manuscript has interest pending some changes that must be done.

First of all, a grahical Abstract Figure must be added and will help in the understanding of the whole review.

Then, when talking of migration, in the Introduction, the Author does not take into consideration that some studies on explant-derived DPSCs have been published some years ago (see the first one by Spath et al 2010 Jun;14(6B):1635-44 and others afterwards). In addition, some other articles have been published on the DPSCs grafting into humans for bone regeneration and this leads to homing.

Then, regarding their role of mechanosensors , YAP/TAZ involvement in DPSCs has been recently  highlighted in literature (see 2021 Oct 26;10(11):2899 and others).

Some citations are redundant and should be erased, mainly when more than one on the same topic.

Review well written, showing only some minor English mistakes.

The quality of the Scientific English is good.

Author Response

This review focuses on specific capabilities of DPSCs.

Albeit the number of reviews on these stem cells are in numerous, this manuscript has interest pending some changes that must be done.

I would like to thank the referee for taking the time to read the review. I hope these changes will satisfy the referee and make the manuscript suitable for publication.

  1. R) First of all, a grahical Abstract Figure must be added and will help in the understanding of the whole review.
  2. A) I thank the reviewer for the suggestion. A graphical abstract has been added in the revised version.
  3. R) Then, when talking of migration, in the Introduction, the Author does not take into consideration that some studies on explant-derived DPSCs have been published some years ago (see the first one by Spath et al 2010 Jun;14(6B):1635-44 and others afterwards). In addition, some other articles have been published on the DPSCs grafting into humans for bone regeneration and this leads to homing.
  4. A) I'm not sure I understood what the referee meant. In the introduction I described in general DPSCs potentiality and then I dealt with in more detail in the following paragraphs. In particular, the migration modalities used by the DPSCs have been described in the “migration” paragraph, and the stimuli in the “the DPSCs fate” paragraph. Moreover, in the article by Spath et al. (2010) suggested by the referee, the authors do not mention migration and the method described for obtaining the cells is not really a classic explantation method. In fact, the authors digested the pulp fragments with trypsin for 5 minutes discarding all obtained cells, and then allowed them to grow on a plate like an explant. I agree with the referee that some other articles have been published on DPSCs grafting as for example Fujii et al. (2018) who reported that hDPSC sheets were implanted in mice calvaria defects and promote bone regeneration. As suggested, in the introduction I added a reference (Fujii 2018) that analyzes hDPSCs in bone regeneration.
  5. R) Then, regarding their role of mechanosensors, YAP/TAZ involvement in DPSCs has been recently highlighted in literature (see 2021 Oct 26;10(11):2899 and others).
  6. A) As suggested by the referee I added the article by La Noce et al. 2021 in the “Mechanosensing and Mechanotransduction” paragraph.
  7. R) Some citations are redundant and should be erased, mainly when more than one on the same topic.
  8. A) As suggested by the referee I checked for redundant references.

Reviewer 3 Report

Dear author,

the article is interesting but it shows a lot of contrasting proofs and ideas.

In my opinion, the author needs to conceptualize the review in a different manner and based on more similar data.

For that general reason I think the article is not ready for publication.

1)       First of all, in my opinion, the analysis and comparison of the fate of DPSC must be done on the same cellular population. It would be better to compare articles with the same sorted DPSCs

2)       The author supposes that:

“The decision between renewal and migration/differentiation of DPSCs depends on interactions with stromal cells and ECMs in the niches. Different niches possess different DPSCs.

That's a really important variable. Different niches have different migratory capacities, hence different cells.

Comparisons of DPSC migration across different niches are not possible.

It would be preferable to compare the DPSC migration potential achieved by the same niches.

Additionally, isolation methods cannot be a different source of DPSC. the author reports this sentence

“Therefore, the two isolation methods yield different subpopulations of cells, even 101 though DPSCs appear to show the same capabilities in their proliferative and trilineage 102 differentiation potentials [15,17,18]”

Comparison of literature based on a single extraction method and homogeneous cell population is recommended.

3)       The migration strategies discussed in paragraph 3 are primarily based on the general literature on mesenchymal stem cells and are not unique to DPSC.

4)       Mechanosensing and mechanotransduction are stimuli for the cell migration.

It should be analyzed in conjunction with tissue damage, which is the most important stimulus for migration.

I believe this subsection could constitute an independent review.

The quality of  English is good

Author Response

Dear author,

the article is interesting but it shows a lot of contrasting proofs and ideas.

In my opinion, the author needs to conceptualize the review in a different manner and based on more similar data.

For that general reason I think the article is not ready for publication.

I would like to thank the referee for taking the time to read the review. The points she/he makes are important. I apologize for the confusion. However, I don't think conflicting ideas are expressed, although there are several findings in the review. The point-by-point answers follow. I hope these changes will satisfy the referee and make the manuscript suitable for publication.

  1. A) The aim of the review is to deal with the migration of DPSCs in relation to the environmental stimuli that can promote it. The environmental cues taken into consideration are tissue damage and mechanical stimuli. The effects of these environmental cues on migration and proliferation/differentiation have been addressed. For a better understanding I added a graphical abstract in the revised version of the manuscript.

1)       First of all, in my opinion, the analysis and comparison of the fate of DPSC must be done on the same cellular population. It would be better to compare articles with the same sorted DPSCs

  1. A) There are two main methods to purify DPSCs from the tooth: enzymatic method, explant method. The enzymatic method produces a heterogeneous population of cells (mesenchymal stem cells from different niches, epithelial cells, perivascular cells, lymphocytes). Usually, this cell population is then selected for the presence of specific stem cell markers and subsequent analyzes are performed on a sub-population of DPSCs. The explant method produces a more homogeneous cell population, due to the leakage of stem cells present in different niches at different times, and to the presence of other non-stem cells which are lost with subsequent passages. However, also this population of cells can be pre-selected for the expression of specific markers, which are functional for subsequent studies. Table 1 has been drawn precisely to be able to compare with which method the DPSCs were obtained. The articles that analyze the migration are not many, among them, those that use the explant method, are a minority, but they are an enrichment of information compared to the other methods since explant-derived DPSCs could have superior migratory capabilities as the isolation principle would favor the selection of cells capable of migration (Bao Ha et al., 2011; Priya et al., 2014). However, in the light of the referee's suggestions where it is possible to compare the results with DPSCs obtained with the same purification method, this has been done in the review.

342-349 Other studies on Exos-DPSCs have used the enzymatic purification method to recover stem cells. In this regard, DPSC-Exos promoted mice cutaneous wound healing-related biological processes, such as the positive regulation of cell motility, migration, proliferation, vasculature development and angiogenesis [Zhou 2022]. Exos-DPSCs obtained under hypoxic conditions (Hypo-Exos) demonstrated that LOXL2, an enzyme that catalyzes the crosslinking of elastin and collagen, is a key molecule in mediating the angiogenic effect by promoting migration and proliferation [Li 2023].

2)       The author supposes that:

“The decision between renewal and migration/differentiation of DPSCs depends on interactions with stromal cells and ECMs in the niches. Different niches possess different DPSCs.

  1. A) This is not my supposing, but the results of other authors (see Lizier 2012, Mitsiadis 2017).
  2. R) That's a really important variable. Different niches have different migratory capacities, hence different cells.
  3. A) Indeed see Bao Ha et al., 2011; Priya et al., 2014; Lizier et al. 2012, Mitsiadis et al. 2017
  4. R) Comparisons of DPSC migration across different niches are not possible.
  5. R) It would be preferable to compare the DPSC migration potential achieved by the same niches.
  6. A) With the enzymatic method it is not possible to recover cells from a single niche, because after enzymatic digestion there is a mixture of cells, and moreover not all the tissue is digested; while instead with the explant method, theoretically, cells from different niches can be physically separated by using successive waves of cells released to recover them (see Vikrant R. Patil 2018).
  7. R) Additionally, isolation methods cannot be a different source of DPSC. the author reports this sentence.

“Therefore, the two isolation methods yield different subpopulations of cells, even 101 though DPSCs appear to show the same capabilities in their proliferative and trilineage 102 differentiation potentials [15,17,18]”

  1. A) I'm sorry there was a misunderstanding. I meant that the enzymatic method produces a heterogeneous population of cells with stem cells from various niches in a mixture, and epithelial cells, stromal cells, perivascular cells, etc.; while instead the explant method produces a more homogeneous population of stem cells (subsequent waves see Vikrant R. Patil 2018) and other cells that do not divide, undergo apoptosis, and are lost in the first passages in vitro. However, DPSCs when not pre-selecting for specific markers expression, showed the same trilineage differentiation potential regardless of the purification method (Huang 2006, Bakopoulou 2011, Hilkens 2013, Kok 2022).

To avoid confusion, I made the following changes throughout the manuscript:

86-89 The harvested cells are dispersed in the medium, plated and left to proliferate and in turn, they form colonies of heterogeneous cells including stem cells from various niches in a mixture, and epithelial cells, stromal cells, perivascular cells, etc. [10] (Fig1A).

102-105 Although, DPSCs obtained by the explant method produce a more homogeneous population (subsequent waves see Vikrant R. Patil 2018), unattached and adherent cells, different from DPSCs, are present in the culture [15].

109-112 Therefore, the two isolation methods yield different subpopulations of cells, even though regardless of the recovery method, DPSCs when not pre-selecting for specific markers expression, showed the same trilineage differentiation potential DPSCs [15,17,18].

  1. R) Comparison of literature based on a single extraction method and homogeneous cell population is recommended.
  2. A) There is a large numerical difference between the published works using the enzymatic method to obtain DPSCs and those using the explant method (9:1 enzymatic/explant). Most of the works using DPSCs obtained with an enzymatic method are pre-selected based on the expression of specific groups of markers useful for the study, especially proliferation and differentiation (STRO-1, CD146, CD271, Oct4, Nanog, ecc.). Furthermore, very often DPSCs selected for the same markers are cultured in different culture media, which prevents comparison of the results. In addition, the enzymatic method does not allow to evaluate the migration capacity of the naïve cells, while instead the explant method allows us to observe the naïve cells (Vikrant R. Patil 2018). Finally, few works compare the two methods to purify DPSCs evaluating proliferation and differentiation, but not migration (Huang, 2007, Lizier 2012, Hilkens 2013). Therefore, comparing the literature, using a single extraction method as a selection criterion, or a homogeneous cell population, always grown under the same conditions, to study migration is hard. The collection of studies on DPSCs obtained with the two methods of separation offers important information that does not overlap, does not contrast, but rather complements each other and makes the overall vision clearer.

3)       The migration strategies discussed in paragraph 3 are primarily based on the general literature on mesenchymal stem cells and are not unique to DPSC.

  1. A) DPSCs are mesenchymal stem cells and for this reason they used all the mechanisms described. Indeed, in the same paragraphs the works that have demonstrated the migration modalities used by the DPSCs (lamellipodia, filopodia, SF, blebbing) have been cited. The only modality, cited in the paragraph 3, not yet demonstrated for DPSCs is the use of migrasomes, which is in fact mentioned as a hypothesis in the paragraph of the conclusions.

4)       Mechanosensing and mechanotransduction are stimuli for the cell migration.

It should be analyzed in conjunction with tissue damage, which is the most important stimulus for migration.

I believe this subsection could constitute an independent review.

  1. A) Mechanical stimuli and tissue damage act as main signals for DPSC migration. Mechanosensing and mechanotransduction are mechanisms by which cells sense the extracellular environment, in particular mechanical stimuli, and convert them into intracellular biological signals. In this regard, new discoveries on signaling (PIEZO1-ATP, YAP/TAZ) in DPSCs have only recently been made (2017, 2018, 2020, 2022). Therefore, it seemed to me interesting to describe and deepen the mechanisms in a separate paragraph (Mechanosensing and Mechanotransduction), leaving the description of the effects of the stimuli (mechanical and biochemical) in relation to migration, proliferation/differentiation in the paragraph "DPSC fate".

Round 2

Reviewer 3 Report

The paper as improved.

Now is acceptable for publication

The paper as improved.

Now is acceptable for publication in the present form

Author Response

Dear reviewer,

thank you for the suggestion. The moderate editing of English will be done by MDPI.